# Ultrafast hole transfer mediated by polaron pairs in all-polymer photovoltaic blends

Rui Wang[1], Yao Yao[2], Chunfeng Zhang[1], Yindong Zhang[1], Haijun Bin[3], Lingwei Xue[3]
Zhi-Guo Zhang[3,4], Xiaoyu Xie[5], Haibo Ma[5], Xiaoyong Wang[1], Yongfang Li[3] & Min Xiao[1,6]

The charge separation yield at a bulk heterojunction sets the upper efficiency limit of an organic solar cell. Ultrafast charge transfer processes in polymer/fullerene blends have been intensively studied but much less is known about these processes in all-polymer systems. Here, we show that interfacial charge separation can occur through a polaron pair-derived hole transfer process in all-polymer photovoltaic blends, which is a fundamentally different mechanism compared to the exciton-dominated pathway in the polymer/fullerene blends. By utilizing ultrafast optical measurements, we have clearly identified an ultrafast hole transfer process with a lifetime of about 3 ps mediated by photo-excited polaron pairs which has a markedly high quantum efficiency of about 97%. Spectroscopic data show that excitons act as spectators during the efficient hole transfer process. Our findings suggest an alternative route to improve the efficiency of all-polymer solar devices by manipulating polaron pairs.

[1] National Laboratory of Solid State Microstructures, School of Physics, and Collaborative Innovation Center of Advanced Microstructures, Nanjing University, Nanjing 210093, China. [2] Department of Physics and State Key Laboratory of Luminescent Materials and Devices, South China University of Technology, Guangzhou 510640, China. [3] Beijing National Laboratory for Molecular Sciences, CAS Key Laboratory of Organic Solids, Institute of Chemistry, Chinese Academy of Sciences, Beijing 100190, China. [4] Beijing Advanced Innovation Center for Soft Matter Science and Enginnering, College of Materials Science and Engineering, Beijing University of Chemical Technology, Beijing 100029, China. [5] School of Chemistry and Chemical Engineering, Nanjing University, Nanjing 210093, China. [6] Department of Physics, University of Arkansas, Fayetteville, Arkansas 72701, USA. These authors contributed equally: Rui Wang, Yao Yao. Correspondence and requests for materials should be addressed to C.Z. (email: cfzhang@nju.edu.cn) or to Z.-G.Z. (email: zgzhangwhu@iccas.ac.cn) or to M.X. (email: mxiao@uark.edu)

The performance of an organic photovoltaic (OPV) device relies on the efficiency of charge separation at its donor/acceptor interface[1–14]. In the well-studied model system of polymer/fullerene blends, charge separation is mainly enabled by electron transfer from polymer donors to fullerene acceptors[1–16]. The process of electron transfer dissociates the photo-excited Frenkel exciton (EX) to form an interfacial charge-transfer (xCT) state, leaving an electron on the fullerene acceptor and a hole on the polymer donor[1]. The xCT state is further separated into a charge-separated (CS) state of free charges for photocurrent generation. In addition to such an EX-mediated channel, another intra-moiety state of charge-neutral excitations with weak interaction between spatially separated electrons and holes, known as the intra-moiety polaron pairs (iPPs), has also been observed upon optical excitation in many polymers[16–26]. The iPP-liked states are generated over time from EX dissociation[22,27] or simultaneously with the formation of EXs[19,21]. It has been an enduring controversy whether such intra-moiety PPs (iPPs) make an important contribution to charge separation in OPV blends[16,23,26,28–30]. In principle, the weakly bound character of iPP-states can be regarded as precursors of free charges for photocurrent generation with small energy loss[19,30]. However, some iPP-liked states in polymers may undergo germinate recombination which competes with free charge generation[16,22,23].

Recently, all-polymer OPV devices have attracted a rapidly growing interest because of their mechanical robustness, morphological stability, and superior flexibility[31–36]. In spite of a marked improvement of power-conversion efficiency (PCE) from 2 to 9% in the last few years[31,36–41], the performances of all-polymer devices still lag far behind the devices using small-molecule acceptors (with PCE up to above 14%)[42–46]. To narrow this PCE gap, it is essential to elucidate the mechanisms of charge generation and transport in all-polymer OPV blends, which have been much less investigated[47–50] in comparison to the intensive studies on polymer/fullerene blends. Apparently, iPPs may be excited in both polymeric donors and acceptors, so the contribution of iPPs to charge separation can be expected to be more substantial in all-polymer blends than in polymer/fullerene blends. However, the role of iPPs in the charge generation in all-polymer OPV blends remains unexplored.

In this context, here we focus on the dynamics of charge separation in an all-polymer OPV blend consisting of a commercial acceptor of naphthalene diimide-bithiophene (N2200)[51], which has been most widely used in high-performance all-polymer devices[31,35–39,41], and a polymer donor of benzodithiophene-alt-benzotriazole copolymers (J51) having absorption coverage commentary to that of N2200[41]. The typical PCE of devices with such J51/N2200 blends (8.3%)[41] is among the highest values reported so far for all-polymer devices[36]. We have observed compelling evidences indicating that the hole transfer process from N2200 to J51 is triggered by the photo-excited iPPs rather than the photo-excited EXs as commonly considered in polymer/fullerene blends. Using transient absorption (TA) spectroscopy, we clearly identify an ultrafast hole transfer process (about 3 ps) with a quantum efficiency up to about 97% from the polymer acceptor to the polymer donor mediated by iPPs, while the photo-induced EXs act as spectators during this efficient hole transfer process. In addition, the iPP-mediated hole transfer process has been identified in other all-polymer blends with donor J51 and three different polymer acceptors, implying that it is a quite general pathway of charge generation in all-polymer OPV systems. The identification of such an iPP-mediated charge separation pathway suggests that it is possible to further improve the efficiency of all-polymer OPV devices by manipulating the generation dynamics of iPPs.

## Results

**Scenario of the iPP-mediated hole transfer process.** In certain conjugated polymers, primary photo-excitation generates iPPs together with EXs within the time resolution of the experiments (typically tens of femtoseconds)[16–21,52,53]. In general, the generation yields of iPPs in donor–acceptor copolymers are much higher than those in homopolymers[19,23]. In the framework of Su–Schrieffer–Heeger (SSH) theory[54], EXs and iPPs are produced in deformational potential valleys induced by self-trapping of electrons and/or holes surrounded by soft vibrational modes. Because of the longer distance between positively and negatively charged polarons, the Coulomb attraction force is weaker for the iPP state in comparison with that for EXs. In consequence, the recombination rates of those iPP states are generally slower than those of EXs[17,19,20,52], and the iPP states may be more easily dissociated into the interfacial separated state of free charges (x(P + P)) than the EXs. Figure 1a schematically depicts the possible processes of hole transfer starting from photo-excited EXs and iPPs. The EX-mediated channel for hole transfer can be understood by analogy with the well-studied electron transfer process in polymer/fullerene systems. Free charges are generated in two steps: EXs → xCT states → CS states (Process 1, Fig. 1a)[1,6,7]. Alternatively, the iPP-mediated hole transfer (Process 2, Fig. 1a) may be directly separated into the x(P + P) state because the iPPs are weakly bound.

**Device performance.** In this study, the commercial polymer acceptor of N2200 (Fig. 1b)[51] was chosen because of its superior performance in OPV devices. The implementation of N2200 has recently stimulated a remarkable improvement in the efficiency of all-polymer devices[31,35,37–39,41]. The polymer donor of J51 was designed and synthesized previously (Fig. 1b) having absorption coverage complementary to that of N2200. The PCEs of the devices with J51/N2200 blend are sensitive to their annealing temperatures. The efficiencies of such devices increase from 5.0 to 8.3% with increasing annealing temperature to 110 °C, but then drops to 5.6% when the annealing temperature is increased further to 200 °C. In OPV blends with fullerene-based acceptors, the contribution of hole transfer from the fullerene acceptor to the polymer donor in the charge generation process is minor because fullerene mainly absorbs light of short wavelength.[7] In all-polymer blends with the acceptor of N2200, the light energy harvested by the polymer acceptor can contribute substantially to charge generation through the process of hole transfer (i.e., charge transfer from the electron acceptor to donor) as observed in other systems with non-fullerene small molecule acceptors[7,13,43–46,55,56]. Figure 1c compares the spectrum of the incident photon-to-current efficiency (IPCE) of a device containing a J51/N2200 blend with the absorption spectra of J51 and N2200. The measured spectra indicate that photons absorbed by either the donor or acceptor are efficiently converted to electrical energy in the device, suggesting that the hole transfer from N2200 to J51 also contributes substantially to the charge generation in this device.

**Interfacial hole transfer process.** Although hole transfer contributes markedly to the performance of all-polymer devices, the dynamics of the hole transfer process in all-polymer blends has been rarely studied[50]. The complementary and well-separated absorption spectral coverages of J51 and N2200 polymers make their blends ideal to elucidate the hole transfer dynamics during charge separation. By selectively pumping N2200 at 1.75 eV, we experimentally monitor the dynamics of hole transfer in the J51/N2200 blend films by ultrafast TA measurements. We keep the pump fluence at a low level (2 μJ cm$^{-2}$) to avoid the excitonic

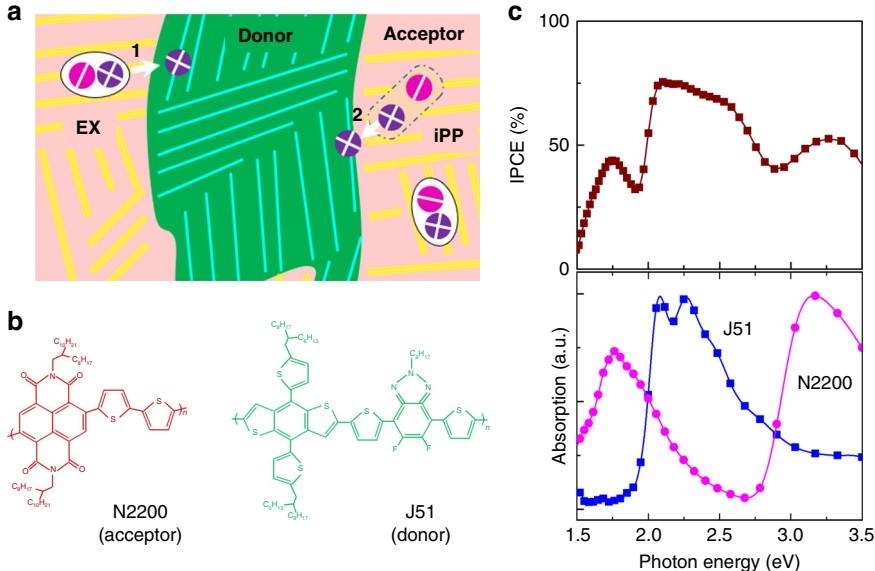

**Fig. 1** Schematics of interfacial hole transfer in OPVs. **a** Two possible channels of hole transfer at a polymer heterojunction: (1) EX-mediated hole transfer and (2) iPP-mediated hole transfer. **b** Chemical structures of the polymer acceptor (N2200) and donor (J51) studied in this work. **c** Absorption spectra of J51 and N2200 (lower panel) are compared with the ICPE spectrum of an optimized device containing a J51/N2200 blend (upper panel)

annihilation effect, which is confirmed by fluence-dependent measurements (Supplementary Figure 1 and Supplementary Note 1). Under such pump conditions, no TA signal is detected from neat films of J51 except for the coherent response at the zero time delay (Supplementary Figure 2), confirming that the primary excitation in the blend films is caused by absorption of N2200 (Supplementary Note 1). Figure 2 shows the TA spectra recorded from a blend film of J51/N2200. The hole transfer process is manifested in the correlated kinetics of ground-state bleaching (GSB) signals of the donor and acceptor (Fig. 2a, b). The GSB signal of polymer N2200 centered at 1.75 eV is simultaneously built up upon photo-excitation. Following the decay of the signal at 1.75 eV, the GSB signal of the donor gradually increases in the range of 2.0–2.4 eV (Fig. 2b, c), suggesting that excitations are transferred from the acceptor (N2200) to the donor (J51). The photon energy of the pump light is below the gap of the donor J51, so exciton transfer from acceptor to donor could be safely excluded. It is reasonable to assign the major origin of excitation transfer to hole transfer considering the staggered band alignment of J51 and N2200[55,56]. Quantitative analysis of the early-state dynamics after the initial coherent response (Supplementary Figure 2b) indicates that the timescale of this hole transfer process is about 3 ps (Fig. 2c), which is consistent with the faster decay of the GSB signal in the blend film than in the neat acceptor film (Fig. 2d).

**Highly efficient iPP-mediated hole transfer process**. The iPP state is peculiar to polymers because their specific molecular structures are different from those of small molecules. However, the role of iPPs during charge separation in all-polymer blends has not been discussed in the limited number of ultrafast spectroscopic studies reported to date[47–50]. We intend to isolate the contributions of EXs and iPPs to the hole transfer process by carefully analyzing the difference between their dynamics in the neat acceptor and blend films. Upon optical excitation, both EXs and iPPs are formed in N2200 which may be uncovered in TA spectra (Supplementary Figure 3). We perform the spectro-electrochemistry measurements[23] to identify the spectral features of iPPs by introducing positive or negative charges into the

polymer N2200 (Supplementary Methods). The experimental results clearly show the increased absorption centered at 1.44 eV and 2.4–2.6 eV for both electron and hole polarons which can be regarded as the absorption features of iPPs (Supplementary Figure 4 and Supplementary Note 1).

Figure 3a displays the TA spectrum of a neat film of N2200 recorded at a time delay of 40 fs with three ESA bands in the probe range (1.0–2.7 eV). The ESA features appear simultaneously upon optical pump within the limit of temporal resolution of the experiment (Supplementary Figure 5), suggesting that they are related to the excited states populated by the initial optical absorption. Referring to the spectrum of polaron absorption (Fig. 3a and Supplementary Figure 4), we can assign the ESA features at 1.44 eV and the visible feature at 2.4–2.6 eV to iPPs. The ESA band centered at 1.12 eV, which is not present in the spectrum of polaron absorption, can be naturally ascribed to the formation of EXs. These assignments are in consistent with a previous study by charge accumulation spectroscopy[49]. The dynamics of the three ESA features are compared in Fig. 3b. The ESA feature at 1.12 eV decays faster than the other two ESA features at 1.44 eV and 2.52 eV with similar dynamic behavior (Fig. 3b), which is consistent with the slower recombination expected for iPPs than that for EXs[17,19,20,52]. TA data for the neat N2200 film can be well reproduced by the global fitting algorithm with the two components of iPPs and EXs having independent temporal dynamics (Supplementary Figure 3). This analysis well supports the above assignments of the ESA features of iPPs and EXs.

The contributions from photo-excited EXs and iPPs to the hole transfer process can be extracted from the difference between their dynamics in the neat acceptor and blend films (Fig. 3c–e). The TA spectrum recorded at a long delay (2000 ps), i.e., the spectroscopic features of the state resulting from hole transfer, exhibits an ESA feature overlapped with the ESA absorption of iPPs near 1.44 eV (Fig. 3a), making it difficult to directly extract the iPP dynamics from the near infrared signal (Fig. 3d). Fortunately, near-zero signals at 1.12 eV and 2.52 eV are observed in the ESA spectrum at 2000 ps, which are helpful to establish the dynamics of EXs and iPPs during the hole transfer process. The photon energy of 2.52 eV is close to the isosbestic point of the

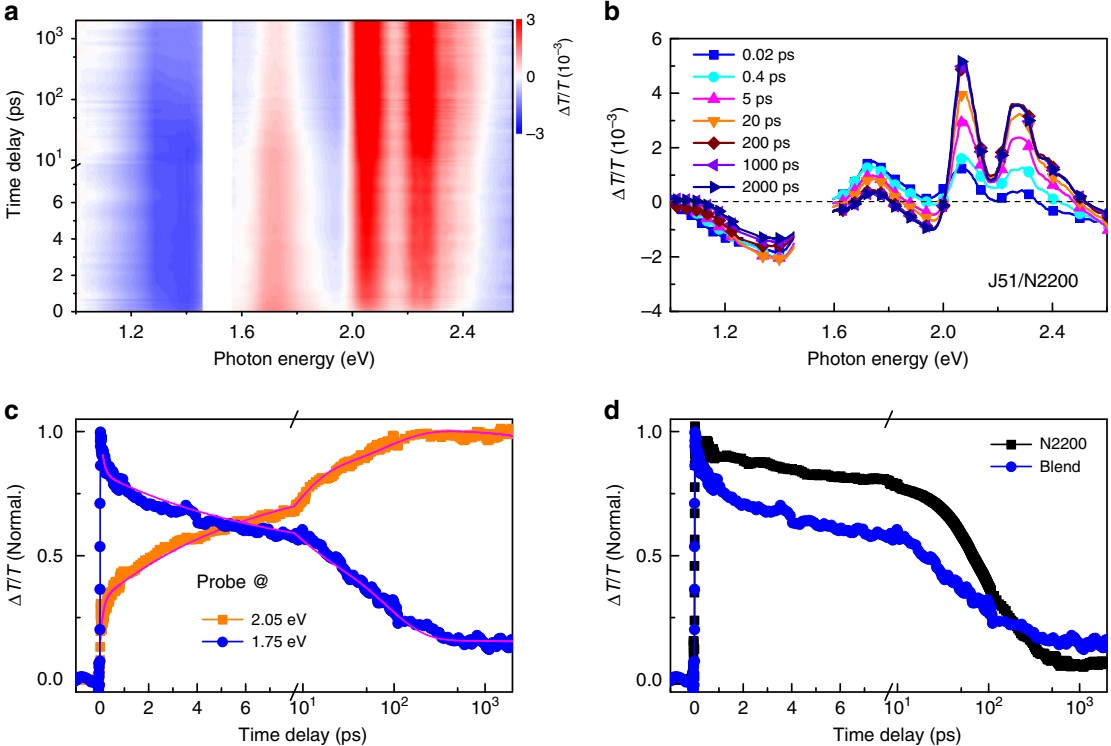

**Fig. 2** Hole transfer dynamics in a J51/N2200 blend film. **a** TA signal recorded from a J51/N2200 blend film with pump photon energy of 1.75 eV. **b** TA spectra from the blend film recorded at different time delays. **c** The GSB signal of the acceptor N2200 at 1.75 eV compared with the GSB signal of the donor J51 at 2.05 eV. The multi-exponential fits show that the hole transfer lifetime is about 3.2 ps. **d** The GSB signal of the acceptor N2200 at 1.75 eV decays much faster in the blend film than in the neat film of N2200

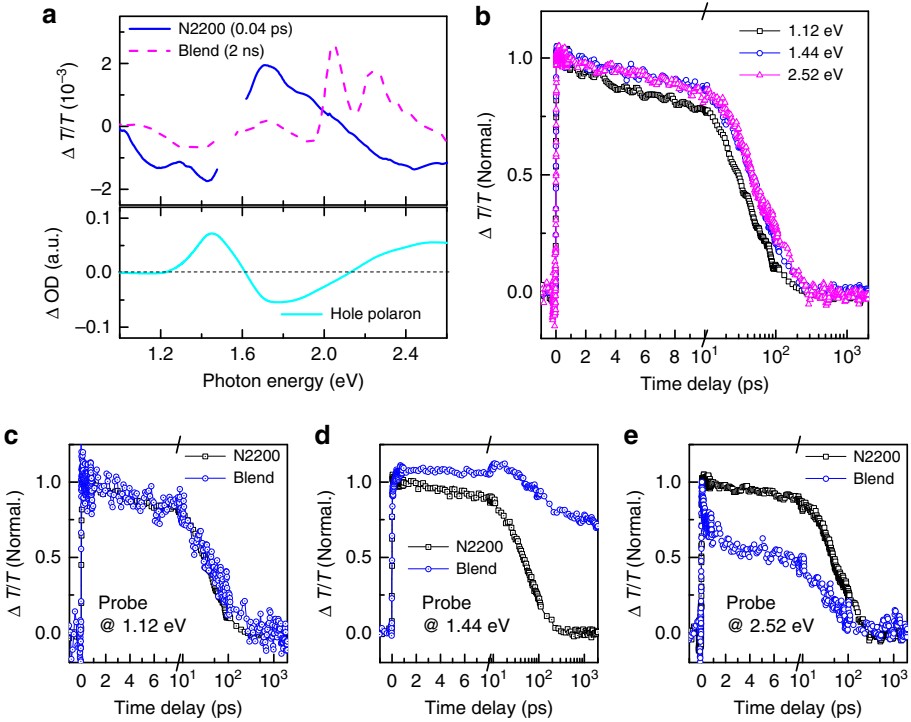

**Fig. 3** Hole transfer mediated by iPPs. **a** A TA spectrum of a neat film N2200 recorded at the time delay of 40 fs is shown in comparison with that of the blend film recorded at 2 ns (upper). One GSB band around 1.75 eV and three ESA bands around 1.12 eV, 1.44 eV, and 2.52 eV are clearly identified. The excitation photon energy is 1.75 eV. The absorption change induced by positive charges in the polymer N2200 characterized by the spectro-electrochemistry measurements shows the absorption features of polarons at 1.44 and 2.5 eV (lower). **b** The decay dynamics of the three ESA bands in the neat N2200 film. The decay dynamics of the neat N2200 and the blend films probed at **c** 1.12 eV, **d** 1.44 eV, and **e** 2.52 eV

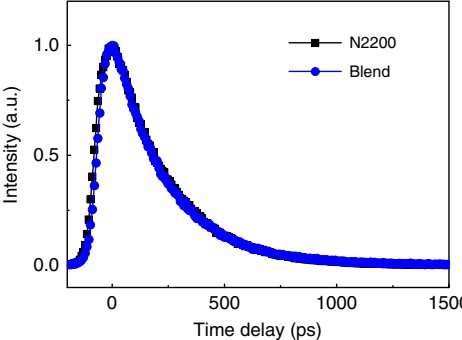

**Fig. 4** Exciton dynamics in the neat acceptor and blend films. TRFL spectra of the neat film of N2200 and the blend film recorded at 1.45 eV, respectively

spectral feature of the state resulting from the hole transfer where the ESA signal and the GSB signals of the resultant state are offset with same amplitudes (Supplementary Figure 6). In this case, the signal probed at 2.52 eV in the blend represents the dynamics of iPPs. As presented in Fig. 3e, the polaron signal probed at 2.52 eV decays much faster in the blend film (about 3 ps) than that in the neat N2200 film (about 107 ps). The lifetime of this ultrafast decay component is consistent with that of the hole transfer manifested in the bleaching signal (Fig. 2c, d), which clearly indicates the involvement of iPPs during the hole transfer process. To further confirm the above assignments, we also try to extract the dynamics of iPPs and EXs during the hole transfer process from ultrafast TA data using the global fitting analysis. As shown in Supplementary Figure 6, the lifetime of the spectral component of iPPs is markedly shortened to about 3 ps in the blend due to hole transfer. By comparing the lifetimes of iPPs in the neat acceptor and blend films, one can estimate that quantum efficiency of hole transfer is about 97% for iPPs, indicating a highly efficient charge separation channel through hole transfer for photo-excited iPPs in N2200.

**The spectator role of EXs.** On the other side, the kinetics of the EX signal probed at 1.12 eV is similar for the neat acceptor and blend films (Fig. 3c), suggesting that the EXs are less relevant to the hole transfer process compared with the case for iPPs. The spectator role of the EXs during the ultrafast hole transfer process is further confirmed by monitoring the fluorescence dynamics of the materials. The measured time-resolved fluorescence (TRFL) spectra show that the decay dynamics of emission from the polymer acceptor in the neat N2200 film is nearly the same as that in the blend film (Fig. 4), which is confirmed by the absence of quenching of fluorescence emission from N2200 (Supplementary Figure 7, Supplementary Note 2). The quantum efficiency of fluorescence emission from polymer N2200 is estimated to be comparable in the neat acceptor and blend films if absorption is calibrated (Supplementary Note 2), confirming that the TRFL dynamics represents the global properties in these samples. Since fluorescence emission mainly originates from EXs, the similar kinetic features in the neat acceptor and blend films, which is consistent with the TA results (Fig. 3c), suggest that the EX dynamics is not significantly affected by the hole transfer process. Notably, the spectator role played by EXs during ultrafast hole transfer is markedly different from that during electron transfer in the J51/N2200 blends (Supplementary Figure 8 and Supplementary Note 3). Photo-excited EXs serve as a major channel for the electron transfer process as evidenced by increased decay rate of fluorescence emission of donor J51 in the blend compared with that in a neat film (Supplementary Figure 7

and Supplementary Note 2). Such a difference is possibly responsible for the lower IPCE of the device in the spectral range of acceptor absorption (Fig. 1c).

We have shown that the hole transfer process in the all-polymer J51/N2200 blends is fundamentally different from the well-established EX-derived channel in typical polymer/fullerene blends. Experimental data strongly support that the hole transfer process is derived from photo-excited iPPs in this all-polymer blend system with a near-unity quantum efficiency. Because the two polarons in the iPP state are weakly bound, free charges (x(P + P)) can be directly produced by the iPP-mediated hole transfer. Such an iPP-mediated channel (i.e., iPP → x(P + P) state) for charge generation, as identified in this work, might hold the key to further improve the light-electricity conversion efficiency in all-polymer devices.

**Morphology effect.** Charge separation happens at the donor: acceptor interface and thus the morphology of the sample plays an important role[1,6,57–59]. The most common methods for morphology engineering include thermal annealing and solvent additive[39,57,60,61]. Here we use thermal annealing method because it mainly affects the phase separation without changing the film thickness and composition ratio. The PCEs of devices with the J51/N2200 blends are dependent on the annealing temperature with an optimized temperature at about 110 °C. Thermal annealing can affect the charge generation, dynamics and transport by changing the crystallizations, phase separations, and molecular orientations in the blends. Here, we focus on the morphology effect on the iPP-mediated hole transfer process and perform TA measurements on the J51/N2200 blend samples before and after thermal annealing at 70 °C, 110 °C, 150 °C, and 200 °C, respectively. The morphologies of these samples are characterized by photo-induced force microscopy (PiFM), which shows the nm-scale spatial patterns of the individual chemical components in their blend films (Supplementary Figure 9).

Figure 5 depicts the correlation between the hole transfer dynamics and the morphology of the blend film. The onset of the bleaching signal probed at 2.05 eV (Fig. 5a) represents the dynamics of hole transfer from N2200 to J51 as discussed above. The dynamics remains nearly unchanged with increasing annealing temperature to 110 °C, whereas annealing at higher temperature decreases the rate of hole transfer (Fig. 5a). In the blend samples, the dynamics of iPPs is sensitive to the annealing temperature (Fig. 5b), whereas the dynamics of EXs probed at 1.12 eV is nearly unchanged (Supplementary Figure 10). The quantum efficiency of iPPs that undergo hole transfer is about 97% in the samples annealed at temperatures below 110 °C, but drops to about 83% in the sample annealed at 200 °C. The dependence of the iPP signal on annealing temperature is correlated with that of the donor bleaching signal (Fig. 5b), confirming that the hole transfer process is triggered by the photo-excited iPPs rather than EXs. The morphologies of these blend samples are compared in Fig. 5d–f and Supplementary Figure 9. In general, the donors and acceptors are mixed in the blends, forming fiber-like nanostructures as revealed in the PiFM images. The phase separations are comparable in the unannealed film (Fig. 5d) and that annealed at 110 °C (Fig. 5e), which explains the similar hole transfer dynamics observed by TA experiments (Fig. 5a, b). In the sample annealed at a higher temperature (Fig. 5f), the domains become well-defined with much larger sizes. The reduced interfacial contacts between donors and acceptors can be ascribed to be the origin of reduced hole transfer rate in the blends annealed at temperatures above 110 °C.

In addition to the hole transfer dynamics, the generation yield of photo-excited iPPs is also sensitive to sample morphology. The

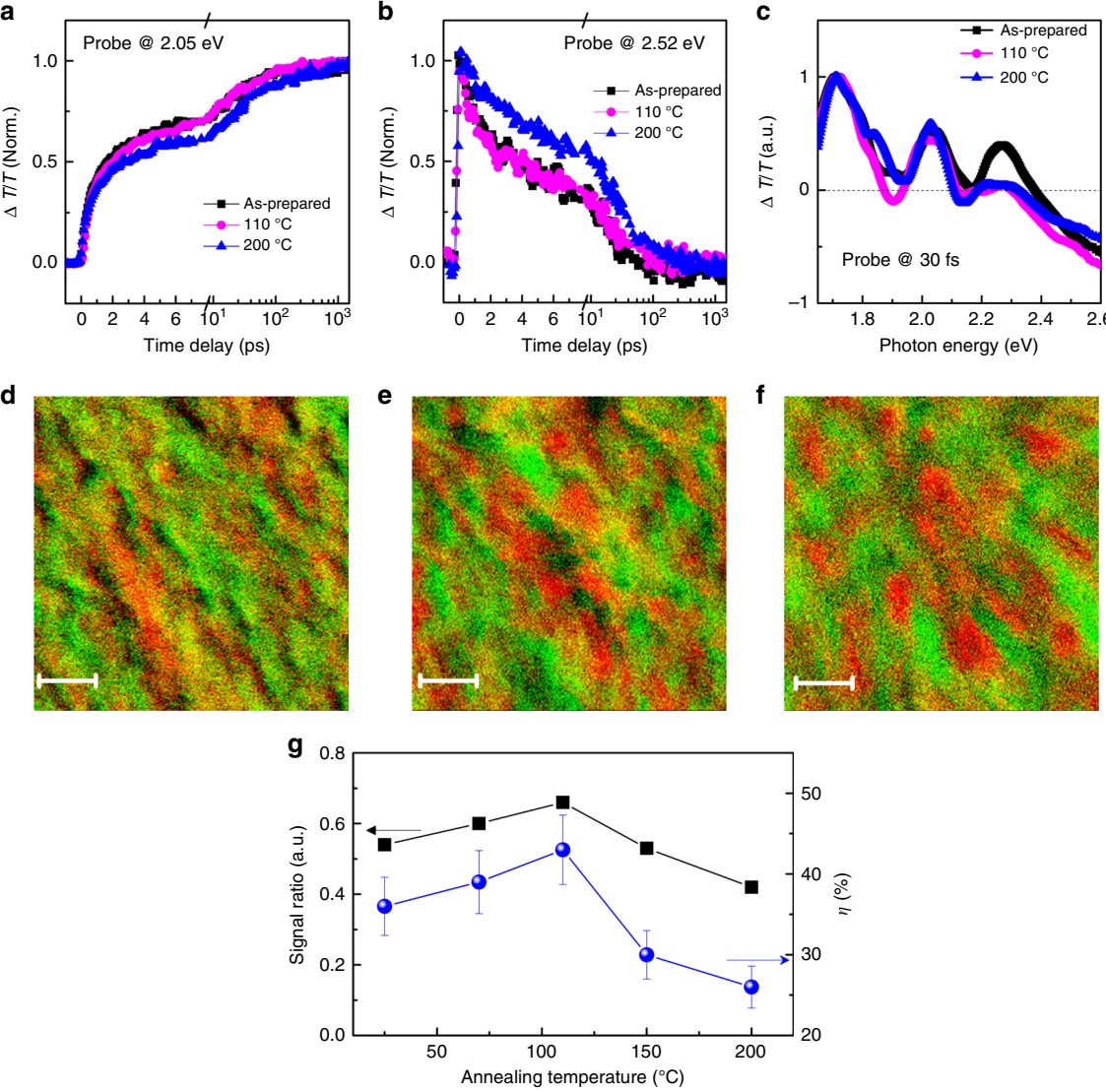

**Fig. 5** Morphology effect on iPP-mediated hole transfer. TA traces probed at **a** 2.05 eV and **b** 2.52 eV recorded from the blend samples before (as-prepared) and after annealing at 110 and 200 °C, respectively. **c** TA spectra recorded at a time delay of 30 fs recorded from the blend samples annealed at different temperatures. The spectra are normalized at to the signal probed at 1.75 eV (Fig. 5c) to compare the relative signal amplitudes of iPPs in different blends. The photon energy of the pump beam is 1.75 eV. PiFM images show the morphologies of the blend samples **d** before (as-prepared) and after annealing at **e** 110 °C and **f** 200 °C, respectively. The characteristic FTIR wavelengths corresponding to the polymer donor (1061 cm$^{-1}$, green) and the polymer acceptor (1712 cm$^{-1}$, red) are selected for better contrast of the PiFM images. The scale bar is 50 nm in **d**, **e**, and **f**. **g** The amplitude ratio between the signals probed at 2.52 eV and 1.75 eV probed at the delay of 30 fs and the estimated efficiency of iPP generation ($\eta$) are plotted versus annealing temperature. Error bars represent standard deviations in the determination of the iPP generation efficiencies obtained by averaging several experiments

GSB signal at 1.75 eV includes contributions from both photo-excited iPPs and EXs, whereas the ESA signal probed at 2.52 eV is mainly contributed by photo-excited iPPs. The amplitude ratio between the signals probed at 2.52 eV and 1.75 eV can thus be used as a qualitative metric reflecting the yield of iPPs generated by primary excitation. As shown in Fig. 5c, g, the generation yield of iPPs increases with increasing annealing temperature below 110 °C, but then drops in the samples annealed at higher temperatures. We quantify the generation yields of iPPs in the samples annealed at different temperatures by using a chemical doping approach (Supplementary Figure 11 and Supplementary Note 4)[19]. The generation yield (Fig. 5f) increases from 36% in the unannealed sample to 43% in the sample annealed at 110 °C and then decreases to 26% in the sample annealed at 200 °C. Such an annealing temperature dependence follows a similar trend to that of the device performance, which is consistent with the

important role played by iPPs in charge generation in the all-polymer blends. While the exact mechanism still requires future in-depth study, such a dependence of iPP generation on annealing temperature is likely related to the polymorphs of N2200 samples as uncovered by Brinkmann et al. by high-resolution transmission electron microscopy[62]. Annealing at low temperature (below 110 °C) may induce molecular reorientation and decrease molecular separation which are beneficial for the generation of iPPs. However, high-temperature annealing changes the form of directional crystallization and disrupts the segregated π-stacking in the films of N2200[62], which lowers the formation yield of iPPs.

The effects of blend morphology on the iPP-mediated hole transfer process are mainly in two folds: (1) The donor/acceptor interfacial area and domain sizes strongly affect the rate of hole transfer; (2) The interchain packing and the segregated π-stacking

strongly affect the yield of iPP photogeneration. Such effects of morphology on charge generation, together with those on charge recombination[1,60,63] and transport[48,60,64,65], may be responsible for the dependence of device PCE on annealing temperature in devices.

## Discussion

We discuss the possible origin of why photo-excited iPPs rather than EXs dominate the hole transfer process in the all-polymer blend. Optical excitation generates both EXs and iPPs simultaneously in the polymer acceptor. The EX-mediated channel for hole transfer needs to overcome a relatively higher barrier than that for iPPs because of EX binding. The energy barrier for the iPP-mediated channel of hole transfer is relatively low, which favors the ultrafast hole transfer observed here. To further examine the possibility of iPP-mediated hole transfer with theory, we use the well-established SSH model[54] to simulate the dynamics of EXs and iPPs at the interface in the J51/N2200 blend using a non-adiabatic dynamics method[66]. The simulation results (including the model parameters) are available in the Supplementary Note 5. The charge density is calculated as a function of delay time and distance from the J51/N2200 interface with an initial incidence of EX or a positive polaron in an iPP state at the donor/acceptor interface (Supplementary Figures 12–15). In the EX case, both the positive (hole) and negative (electron) charges mainly stay at the interface up to 1 ps (Supplementary Figure 15), implying that the exciton is not efficiently dissociated on this timescale. In the iPP case, the positively charged polaron moves from the interface into the donor at a constant speed (Supplementary Figure 15). These theoretical simulations clearly suggest that the ultrafast and efficient charge separation through iPP-mediated hole transfer is possible in the J51/N2200 blends, in good agreement with our experimental observations.

The consistent experimental and theoretical results show that the process of iPP-mediated hole transfer can act as an effective channel for interfacial charge separation in OPV blends, which is likely to be an important factor responsible for the improved efficiency of J51/N2200-based devices. It is also evidenced that the dependences of the iPP yield and hole transfer rate on the annealing temperature (Fig. 5) are correlated with that of the device efficiency. The identification of such an important process has immediate implications to further improve OPV devices. First, it suggests a promising way to optimize the efficiency of charge separation by using this iPP-mediated channel (iPP → x (P + P) state) from polymer acceptors. This process represents a potential advantage of using all-polymer systems, because the formation of iPPs is generally much more efficient in polymers than that in small molecule acceptors[7]. Second, the iPP state has a lower binding energy with a longer lifetime than those of the EX state, which is meaningful when optimizing the photovoltage output of an OPV device[67]. Third, the iPP-mediated channel coexists with the EX-mediated channel for charge separation. In the J51/N2200 system, EXs play a major role in the ultrafast electron transfer process from donor to acceptor (Supplementary Figure 8 and Supplementary Note 4), as evidenced by the strong quenching of donor fluorescence in the blend film (Supplementary Figure 7). To further improve the efficiencies of such all-polymer devices, new strategy is required to facilitate the EX-mediated channel for hole transfer. To fabricate a practical device, it will be important to design a blend system with optimal band alignment and molecular aggregation so that cooperative functions of EX- and iPP-mediated channels can be achieved to optimize charge separation.

The process of iPP-mediated hole transfer is not present in polymer/fullerene systems because iPP generation is inefficient in fullerene (Supplementary Figure 16 and Supplementary Note 6). In principle, iPPs may be heavily involved in the electron transfer process in polymer/fullerene systems[22,24,25,27,68]. However, charge photogeneration in polymer/fullerene systems has been mainly explained by the EX-derived electron transfer process[1–12,15,16] even in the systems with donor–acceptor copolymers where the generation efficiencies of iPPs are expected to be high[2,13,14,68]. In some polymers, iPP-liked states are generated over time from dissociation of EXs which may act as intermediate states for EX-initiated charge separation processes[22,27]. In this picture, the germinate recombination of some iPP-liked states may compete against the generation of free charges[16,22,23]. In all-polymer J51/N2200 blends, the role played by iPPs in the hole transfer process shows substantially differences from that in the EX-initiated electron transfer process. The hole transfer process is triggered by the iPPs instead of the EXs. This divergence is likely related to the different generation dynamics of the iPP states. In the polymer acceptor N2200, the iPPs and EXs are simultaneously generated (Supplementary Figure 5) similar to the results reported by Tautz et al.[19] and De Sio et al.[21]. Coherent vibronic coupling has been proposed to be responsible for the simultaneous formation of iPPs and EXs[21]. The initial optical absorption creates a coherent state composed of vibronically coupled iPPs and EXs which is then converted to iPPs and EXs due to the loss of coherence[21]. After that, the iPPs and excitons have different and uncorrelated dynamics[19]. The generation dynamics of iPPs in N2200 are consistent with the coherent scenario. TA measurements with sub-10 fs resolution (Supplementary Figure 5) show signatures of coherence with oscillatory behaviors in the ESA signals of iPPs and GSB signals at a frequency of about 1450 cm$^{-1}$. These results may explain the hole transfer process mediated independently by iPPs. In addition, the divergent behaviors of iPPs in the polymer/fullerene and all-polymer blends may arise from the differences in the charge withdrawing ability of polymer and fullerene, the localization of the photo-excited states, the involved phonon modes and the blend morphology[69].

We have also studied other blend films with polymer J51 and three different naphthalene diimide-based copolymer acceptors, i.e., P(IDT-NDI), P(TVT-NDI) and P(NDI2DT-T) (Supplementary Figure 17 and Supplementary Note 6). Similar iPP-mediated hole transfer processes have been identified in all these blends, suggesting that our findings are quite general for all-polymer OPV systems.

In summary, our measurements have established that ultrafast hole transfer (about 3 ps) in the all-polymer blends of J51/N2200 is triggered by photo-excited iPPs. Such a charge separation pathway is fundamentally different from the EX-dominated channel in the polymer/fullerene systems. The newly identified hole transfer process mediated by iPPs can be controlled by morphology engineering in the all-polymer blends. Technically, the photogeneration yield of iPPs in copolymers can be improved by engineering chemical structures, separation distances and energy-level arrangements of donor and acceptor units, and interchain π-aggregation[19,20,23]. The generality of such an iPP-mediated hole transfer has been confirmed in multiple all-polymer blends with different polymer acceptors. The results reported in this work may stimulate future efforts to consider the pathway of charge separation mediated by iPPs and make it cooperate with EXs jointly to further increase the efficiency of all-polymer devices.

## Methods

**Sample preparation**. Polymer donor J51[41] and acceptor N2200[51] were synthesized by the methods reported in literature. We fabricated the photovoltaic devices by the procedure described in Supplementary Methods[41]. The IPCE spectrum (Fig. 1c) was measured by a Solar Cell Spectral Response Measurement System QE-R3−011

(Enli Technology Co., Ltd., Taiwan). The light intensity at each wavelength was calibrated with a standard single-crystal silicon photovoltaic cell. The film samples for TA measurements were prepared by spin coating. The blend samples were made with 2:1 mass ratio of donor to acceptor. The thicknesses of film samples were about 100 nm. The samples were annealed in Argon atmosphere at different temperatures for 10 min. The morphologies of the samples were characterized by PiFM (Vista-IR, Molecular Vista).

**Optical characterizations**. A Ti:sapphire regenerative amplifier (Libra, Coherent Inc.) was used for TA spectroscopy. A home-built non-collinear optical parametric amplifier pumped by the regenerative amplifier was used to generate the pump beam with photon energy centered at 1.75 eV or 2.05 eV for excitation of acceptor and donor. The pump fluence was kept at a level below $2 \, \mu J \, cm^{-2}$. In this regime, the effect of exciton–exciton annihilation can be largely avoided as confirmed by fluence-dependent measurements (Supplementary Figure 1). The probe beam was a broadband supercontinuum light source generated by focusing a small portion of the femtosecond laser beam onto either a 2 mm-thick sapphire plate for the visible range or a 5 mm-thick plate of Yttrium aluminum garnet for the near-infrared range. The supercontinuum probe was compressed with chirp mirrors. The temporal resolution was better than 20 fs in the visible range or 30 fs in the near infrared range. The TA signal was then analyzed by a silicon charge-coupled device (CCD; S11071, Hamamatsu) for visible range or an InGaAs CCD (G11608, Hamamatsu) for infrared range mounted on a monochrometer (Acton 2358, Princeton Instrument) enabled by a custom-built control board from Entwicklungsbuero Stresing. The signal-to-noise ratio in differential transmission was about $10^{-5}$ after accumulating and averaging 2500 pump-on and pump-off shots for each data point. The angle between the polarized pump and probe beams was set at the magic angle. TRFL spectra were recorded with the technique of time-correlated single-photon counting by an avalanche photodiode with a temporal resolution of about 50 ps. During optical measurements, the samples were placed in nitrogen atmosphere to avoid the photodegradation.

Spectro-electrochemistry and chemical doping measurements were conducted to check the absorption features of iPPs in the ultraviolet-visible-near infrared spectral range. For spectro-electrochemistry measurements, positive or negative charges were introduced into polymer N2200 films by applying different voltage versus Ag/AgCl[23]. The differences in the absorption spectra were measured in a same sample before and after charging to extract the absorption spectrum caused by polarons. For quantitative calibration of the absorption coefficient of polarons, we followed the literature approach[19,70] and conducted the chemical doping experiments in a solution sample using a dopant of tris-(pentafluorophenyl)borane (see Supplementary Methods for details).

**Theoretical modeling**. The one-dimensional SSH model was considered with a unit of thiophene acting as the lattice site. Parameters, including the electronic coupling, the primary vibrational mode, and the vibronic coupling, were all calculated by the first-principle computations. A non-adiabatic dynamic simulation was performed for both EX and iPP configurations. More details are available in Supplementary Note 5.

## Data availability
The experiment data that support the findings of this study are available from the corresponding author upon reasonable request.

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

## Acknowledgements

This work is supported by the National Key R&D Program of China (Grant No. 2017YFA0303700 and 2018YFA0209101), the National Science Foundation of China (Grant No. 21873047, 91850105, 11574140, 91833305, 11621091, 11574052, 51722308, and 91633301), Jiangsu Provincial Funds for Distinguished Young Scientists (BK20160019), the Priority Academic Program Development of Jiangsu Higher Education Institutions (PAPD) and the Fundamental Research Funds for the Central University. C.Z. acknowledges the financial support from Tang Scholar program. The authors acknowledge Liang Gao, Lian Zhong and Jia Yao for their help in device fabrication and characterization, and Dr. Xuewei Wu for his technical assistance.

## Author contributions

C.Z. and M.X. initiated the project. R.W. and Y.Z. performed optical experiments. H.B., L.X., Z.-G.Z. and Y.L. prepared the samples and characterized devices. R.W., C.Z. and X.W. analyzed the data. Y.Y., X.X. and H.M. performed the numerical simulation. R.W., C.Z. and M.X. co-wrote the manuscript with help from all other authors.

## Additional information

**Competing interests:** The authors declare no competing interests.

