## [Peer Review File · Nature Communications]

Reviewers' comments:

Reviewer #1 (Remarks to the Author):

Comments on "Ultrafast hole transfer mediated by polaron pairs in an all-polymer photovoltaic blend" by R. Wang et al.

Claims:

1. "Here, we show that interfacial charge separation can occur through a polaron pair-derived hole transfer process in an all-polymer photovoltaic blend" – I am not convinced they show this.
2. "we have clearly identified an ultrafast hole transfer process with a lifetime of ~3 ps mediated by photo-excited polaron pairs which has a markedly high quantum efficiency of ~ 97%." Again, the 3 ps hole transfer is clear, but assignment to the iPPs is not.
3. "excitons act as spectators during the efficient hole transfer process" I have concerns that this conclusion is based on sample heterogeneity.

The authors report keeping pump fluences around 2 uJ/cm² to avoid annihilation effects. Did they do a fluence dependence to characterize where it kicks in? Did they do the measurement on vacuum, under nitrogen or on encapsulated samples? Was photo degradation observed under any conditions?

I believe there is a conceptual error in the argument assigning the ESA bands. The premise 1 used by the authors is that "The recombination of iPPs is slower than that of EXs". If excitons are really recombining faster, then how is that consistent with the data from figure 4, which shows exciton emission for over 1 ns? Furthermore, exciton recombination implies not only a decay in the ESA and stimulated emission, but also in the ground state bleach. So the assignment of the band at 1.12 as the ESA of the excitons seems to be plagued by flawed logic, with obvious sanity checks (global fit of the TA data showing if the faster decay is also present in the bleach signals) not being performed.

In line 167 the authors say "The ESA peaked at ~ 2.6 eV closed to the polaron absorption in CAS spectrum can be ascribed to the ESA feature of iPPs with a relatively long lifetime." I find this very unconvincing, as the traces in Figure 3b barely show differences. To me, the tiny differences cannot be used to assign the 2.6 eV ESA band to iPPs. Why are they not intrinsic ESA from the excitons, for instance? Perhaps the authors are correct, as I am not familiar to the CAS studies they cite, but the manuscript as currently presented does not substantiate the assignment of the 3 ESA bands in a sufficiently clear way. Further, in Figure 3b the normalization point of the 1.12 eV trace seems to be lower than that of the 2.6 eV trace, possibly explaining why the black trace lies slightly below the red and black. In summary, to me it seems more likely that all three traces in that figure correspond to one species only.

Further, in Figure 3e the authors compare dynamics of blend and acceptor films at 2.52 eV. That is an unfortunate choice, because 2.52 eV is very close to the point of zero signal in the blend, consisting of a sum of ESA contributions with bleach from the donor.

Could Figure 3c really be showing that the 1.2 band is due to defects in the acceptor structure?

Another point that the authors do not address is that from the very first TA map they report of the blend (400 fs), a bleach of the donor is present. As the authors showed that there is no direct excitation of the donor with their 1.75 eV pump, this implies that substantial charge separation has already occurred by 400 fs!

Further, the data from Figure 4 seems indeed strange. Are the authors sure the signal they measure in the blend is not arising from a part of the film where mostly the acceptor is present? My feeling is that the data from Figure 4 might be representing inhomogeneity of the samples more than a fundamental process occurring dominantly. The hole transfer takes place on a timescale below the temporal resolution, so the authors only see signal of isolated acceptor domains that emit like the pristine. What determines the ratio of excitons and iPPs? My understanding is that iPPs do not absorb light directly, so if they are formed over time, why are there any excitons left behind to emit? Could the authors perform electroluminescence measurements or PL quantum yield to address this?

Finally, the authors do not perform similar experiments pumping the donor to characterize the electron transfer dynamics. From their figure 1c it seems like the most important contribution to device performance comes from electron transfer, and this data seems like a natural complement.

In summary, I have great suspicion regarding the assignment of the iPP and exciton spectral features, which is the foundation for the discussion that follows. I also think that the paper is not very well written, lacking objectiveness and conciseness. The authors are repetitive a few times, wasting space that could be devoted to clearer explanations on more critical points. For instance, if the authors want to publish in a broad readership journal, at least a basic discussion to help readers from nearby fields to orient themselves on the iPP concept along with literature citations to orient readers. Perhaps this is already provided, but certainly not in a well organized and fluid way.

I think the paper cannot be published anywhere as it is, and could be reconsidered at Nature Communications following major revisions which address the points above.

Reviewer #2 (Remarks to the Author):

The manuscript intended to suggest a new mechanism in all polymer solar cell which is distinctively different than polymer/PCBM solar cell. They investigated role of so called intra-moiety polaron pairs (iPPs) and concluded that these iPPs play an important role that was not present in polymer/fullerene solar cells. I am afraid that this conclusion is totally wrong and the authors lack the knowledge in fundamental photophysics of conjugated polymers. In particular, they have not fully understood the mechanisms revealed so far on so called charge transfer polymers or donor-acceptor polymers or low band gap polymers. There have been numerous literature since ~2005 (2005 PRL Friend and coworkers, 2008 JACS Durrant and coworkers, 2012 JACS Rolczynski et al., and many others) to show that the built-in charge transfer nature within a single polymer as well as their likely self-folding conformation promote the generation of charge transfer (CT) states after exciton splitting where the hole and electron are separated locally which is equivalent as the authors' iPPs. It has been suggested in the literature that the nature of such CT or iPP states in these polymers defray the exciton binding energy due to the hole-electron displacement along the polymer chain. Therefore, their work just proved such mechanism because they also used charge transfer polymer as the p-typed material. There is no fundamental difference between the all-polymer solar cells and polymer/fullerene solar cells. Their perceived "new" mechanism is a part of fundamental photophysics in these organic materials. It is my personal view that the main difference between the all polymer solar cell vs. polymer/fullerene solar cell will be their donor-acceptor morphology which have not been fully explored. Based on the above assessment, I disagree with authors view and hence will not recommend its publication in Nature Communication due to the lack of novelty.

Reviewer #3 (Remarks to the Author):

The research topic is very interesting. It disclosed the hole transfer from the acceptor polymer to the donor polymer in all-PSCs is mediated by photo-excited polaron pairs, which is different from the well-established polymer/fullerene system. This study provided a deep understanding of all-polymer system and is meaningful for the further development of all-polymer systems. Few questions need to be addressed before considering for publishing.

1. It will be very interesting to make a direction comparison to the polymer/fullerene system, for instance using J51/PCBM[70].
2. It is hard to draw a general conclusion on all-PSCs based on only one material system. It will make the study more interesting by extending to another polymer acceptor system.
3. When the morphology changes, how it will influence the quantum efficiency of hole transfer? The numbers need to be provided.
4. If the hole transfer is so efficient, how about the internal quantum efficiency? What is the main loss mechanism since the EQE is not so high?

Response to Reviewers' Comments on MS #NCOMMS-18-23985 by Wang *et al*:

We thank the reviewers for their careful reviews of our manuscript and for their constructive comments, which have helped us to further improve our manuscript. In the past months, we have taken additional data, redone some analysis, and modified the presentation in our manuscript to address all the reviewers' comments/questions. We believe that the revised manuscript is much improved from the original version and it can now be recommended for publication in *Nature Communications*. In the following, we present our point-to-point responses to all the reviewers' comments and indicate corresponding changes made in the revised manuscript. The major changes have been highlighted by color in the "revised manuscript with marks".

Response to Reviewer #1's comments:

Reviewer's main comments:

Claims:

- 1. "Here, we show that interfacial charge separation can occur through a polaron pair-derived hole transfer process in an all-polymer photovoltaic blend" – I am not convinced they show this.*
- 2. "we have clearly identified an ultrafast hole transfer process with a lifetime of ~3 ps mediated by photo-excited polaron pairs which has a markedly high quantum efficiency of ~ 97%." Again, the 3 ps hole transfer is clear, but assignment to the iPPs is not.*
- 3. "excitons act as spectators during the efficient hole transfer process" I have concerns that this conclusion is based on sample heterogeneity.*

Response: We appreciate the reviewer's careful reading of the manuscript and the thoughtful comments.

The reviewer's major concern is about the assignment of ESA feature of the intra-moiety polaron pairs (iPPs) and excitons. To address this issue, we have performed additional spectro-electrochemistry and chemical doping measurements (results are shown on Figure R1 below and Supplementary Figure 11 in Supplementary Information (SI)). These approaches have been well established as reliable ways to identify the spectral features of polaron pairs (e.g., Di Nuzzo *et al*, *JPC Lett.* **6**, 1196 (2015); Tautz *et al.*, *Nature Commun.* **3**, 970 (2012)). The new results strongly support the assignment of ESA features at 1.44 and 2.4-2.6 eV to iPPs. No ESA peak of polarons is detected at 1.12 eV, which is consistent with its assignment to excitons. With these new evidences, the reviewer's concerns about the 1st and 2nd claims are well addressed. In addition, we have recorded the TA data with a higher temporal resolution and an improved signal-to-noise ratio. The differences

between the dynamics of the ESA features of iPPs (at 1.44 and 2.4-2.6 eV) and excitons (1.12 eV) are now better resolved (see below response to technical comment #5). With these justified assignments, we can safely conclude that the iPPs mediate a major channel of hole transfer in the polymer blends. For the concern on the 3rd claim, we have followed the reviewer's detailed comment (see below response to technical comment #7) and measured the quantum yield of photoluminescence emission. In the blend and neat acceptor films, the quantum efficiency of PL emission from N2200 is comparable if the absorption is calibrated, which confirms the insignificant role played by excitons during hole transfer process. Sample heterogeneity should not be a major factor.

Figure R1. In-situ spectro-electrochemistry of polymer N2200. (a) Absorption spectra of polymer N2200 recorded before and after introducing positive charges at 0.5 V. (b) The absorption change caused by positive charges showing the absorption bands of hole polarons at 1.44 and 2.4-2.6 eV. (c) Absorption spectra of polymer N2200 recorded before and after introducing negative charges at -1.5 V. (d) The absorption change caused by positive charges showing the absorption bands of electron polarons at 1.44 and 2.4-2.6 eV.

In the revised manuscript, we have made revisions accordingly. For the signal assignments, Figure 3a is modified by including the spectrum of polaron absorption. The 2nd and 3rd paragraphs on Page 8-9 have been rewritten to justify the assignments of iPPs and EXs. The data of spectro-electrochemistry and chemical doping measurements are included in Supplementary Figure 4 & 11. More relevant discussions are included in Supplementary Note 1 & 4. For fluorescence

measurements, “The quantum efficiency of... in these samples” have been included on Page 9. Supplementary Figure 7b have been added. More relevant discussions are included in Supplementary Note 2.

In the following, we answer the reviewer’s detail comments point by point and indicate the relevant changes made in the revised manuscript.

Reviewer’s detail Comment #1:

The authors report keeping pump fluences around $2 \mu\text{J}/\text{cm}^2$ to avoid annihilation effects. Did they do a fluence dependence to characterize where it kicks in? Did they do the measurement on vacuum, under nitrogen or on encapsulated samples? Was photo degradation observed under any conditions?

Response: As shown in Figure R2, we have measured the power-dependent data of N2200 film. The dynamics of the ground state bleaching signal probed at 1.75 eV is nearly independent of the pump fluence in the range $< 5 \mu\text{J}/\text{cm}^2$, confirming that the annihilation effects can be neglected at $2 \mu\text{J}/\text{cm}^2$. During the measurements, the samples were kept in nitrogen atmosphere and almost no photodegradation was detected during the time framework of our measurements. This power-dependent data (Figure R2) are included in the revised SI as new Supplementary Figure 1. The related discussions are included in Methods as “During optical measurement...avoid the photodegradation” in the revised manuscript and Supplementary Note 1 in the revised SI.

Figure R2. Temporal evolution dynamics of the ground-state bleaching signal in polymer N2200 probe at 1.75 eV and recorded under different pump fluences.

Reviewer's Technical Comment #2:

I believe there is a conceptual error in the argument assigning the ESA bands. The premise I used by the authors is that "The recombination of iPPs is slower than that of EXs". If excitons are really recombining faster, then how is that consistent with the data from figure 4, which shows exciton emission for over 1 ns? Furthermore, exciton recombination implies not only a decay in the ESA and stimulated emission, but also in the ground state bleach. So the assignment of the band at 1.12 as the ESA of the excitons seems to be plagued by flawed logic, with obvious sanity checks (global fit of the TA data showing if the faster decay is also present in the bleach signals) not being performed.

Response: With the strong support of our new experimental results (Figure R1), we can safely assign the ESA features of iPPs and excitons without relying on the premise. As shown in the above Figure R1, we performed the spectro-electrochemistry measurements on the sample of polymer N2200 and measured the absorption induced by polaron formation from visible to near infrared spectral range. The results clearly show that the absorption bands of both electron polaron and hole polaron are around 1.44 eV and 2.4–2.6 eV, respectively. The absorption band around 1.12 eV is not observed in the absorption band of either electron or hole polarons, which can be naturally assigned to the photon-induced excitons.

In the original manuscript, TRFL curves in Figure 4 are plotted in a logarithm scale. At 1 ns, the intensity of FL emission has ~ 1 percent left. The PL lifetime is ~ 230 ps. In the TA dynamics, the kinetics of ESA signal of excitons decay faster. The difference is possibly related to the different temporal resolutions of the TA and TRFL measurements and different contributions from emissive and non-emissive excited states to the TA and TRFL signals.

Figure R3. (a) Experimental results and (b) simulation data of TA spectra of the N2200 film. The simulation is based on a global fitting algorithm. The experimental data can be reproduced by assuming two components (of iPPs and EXs) with spectral and temporal characters shown in (c) and (d). The temporal characters are similar to that probed at photon energies of 1.44 eV and 1.12 eV, respectively, as shown in (e).

In the revised manuscript, we show the TA spectra recorded with a higher temporal resolution (< 20 fs in the visible range and < 30 fs in the near infrared range) and an increased signal-to-noise ratio where the differences in the dynamics of iPPs and excitons are better resolved (Figure R3). Following the reviewer’s suggestion, we have performed global fitting to analyze the data as shown in Figure R3. The spectra for both iPPs and EXs show bleach signals at 1.75 eV, confirming the presence of a faster decay component in the ground state bleach signals as noted by the reviewer.

In the revised manuscript, Figure 3a is modified by including the polaron absorption spectrum. The data of spectro-electrochemical and chemical doping measurements are included in Supplementary Figure 4 & 11. The 2nd and 3rd paragraphs on Page 8-9 have been rewritten to justify the assignments of iPPs and EXs. Figure R3 is included as Supplementary Figure 3 in the revised SI. More related discussions are included in Supplementary Note 1 in the revised SI.

Reviewer’s Technical Comment #3:

In line 167 the authors say “The ESA peaked at ~ 2.6 eV closed to the polaron absorption in CAS spectrum can be ascribed to the ESA feature of iPPs with a relatively long lifetime.” I find this very unconvincing, as the traces in Figure 3b barely show differences. To me, the tiny differences cannot be used to assign the 2.6

eV ESA band to iPPs. Why are they not intrinsic ESA from the excitons, for instance? Perhaps the authors are correct, as I am not familiar to the CAS studies they cite, but the manuscript as currently presented does not substantiate the assignment of the 3 ESA bands in a sufficiently clear way. Further, in Figure 3b the normalization point of the 1.12 eV trace seems to be lower than that of the 2.6 eV trace, possibly explaining why the black trace lies slightly below the red and black. In summary, to me it seems more likely that all three traces in that figure correspond to one species only.

Response: We thank the reviewer for pointing out this potential problem. As noted above, we now present the results of additional measurements to justify these assignments. The spectro-electrochemistry and chemical doping measurements have been well established as reliable ways to identify the spectral features of polaron pairs (see e.g., Di Nuzzo et al., JPC Lett. **6**, 1196 (2015); Tautz et al., Nature Commun. **3**, 970 (2012)). In brief, the polarons are introduced by electrochemical treatment or chemical doping. We observed that the absorption bands of the polarons are around 1.44 eV and 2.52 eV but not around ~ 1.12 eV. Those results are consistent with the assignments of spectral features for iPPs and excitons. In the revised manuscript, we present our newly-recorded TA data with a higher temporal resolution (< 20 fs in the visible range and < 30 fs in the near infrared range) and improved signal-to-noise ratio. The differences in the dynamics of these ESA features of iPPs and excitons are now better resolved (see Figure R3 above). We hope that these new results can relieve the reviewer's concern on this issue.

In the revised manuscript Figure 3a is modified by including the polaron absorption spectrum. The data of spectro-electrochemical and chemical doping measurements are included in Supplementary Figure 4 & 11. The 2nd and 3rd paragraphs on Page 8-9 have been rewritten to justify the assignments of iPPs and EXs. More related discussions are included in Supplementary Note 1 in the revised SI. The experimental details are included in Supplementary Methods.

Reviewer's Technical Comment #4:

Further, in Figure 3e the authors compare dynamics of blend and acceptor films at 2.52 eV. That is an unfortunate choice, because 2.52 eV is very close to the point of zero signal in the blend, consisting of a sum of ESA contributions with bleach from the donor.

Response:

Figure R4. Simulated TA data with the global fitting algorithm. The experimental data (Figure 2a) can be well reproduced by the simulation (a) considering three components with the specified (b) spectral and (c) temporal characteristics. In comparison to the features in the neat acceptor film, the two decay components (Spec1 & Spec 2) can be assigned to the iPPs and EXs, respectively.

We choose the signal probed at 2.52 eV in the ESA band of iPPs because it is near the isosbestic point of the spectral feature of the state resulting from the hole transfer. At the isosbestic point, the ESA signal and the GSB signal of the resultant state are offset with same amplitudes, so that the ESA signal probed at 2.52 eV represents the dynamics of iPPs created in N2200 in the blend film. The reviewer's concern on possible contamination of GSB signal can be further addressed if the data probed in a broader band are considered. The faster decay component in the blend is observed in the ESA signals probed with a broader spectral range of 2.4 – 2.6 eV. In the spectral range, the TA signal of resultant state exhibits GSB feature in the probe range < 2.5 eV and ESA feature in the probe range > 2.52 eV. The presence of a faster decay of ESA feature in such a broad spectral range is mainly contributed by the hole transfer process. To further support our assignment, we have performed a simulation of the TA data based on the algorithm of global fitting as shown in Figure R4, which clearly shows that the feature of faster decay in the spectral range 2.4-2.6 eV is related to the iPP-mediated hole transfer.

We included Figure R4 as the new Supplementary Figure 6 and the related

discussion “The photon energy...generated in N2200” and “To further confirm...in the blend” on Page 9-10 in the revised manuscript.

Reviewer’s Technical Comment #5:

Could Figure 3c really be showing that the 1.2 band is due to defects in the acceptor structure?

Response:

Figure R5. Normalized traces of ESA features probed at 1.12 eV and 1.44 eV showing the simultaneous buildup of iPPs and excitons.

This is an insightful question. Of course, it is possible that structure imperfections lead to the localization of the photo-induced excitonic states with the ESA feature at 1.12 eV. These localized excitonic states are less involved in the hole transfer process, which is a possible reason for the inefficient hole transfer observed in the J51/N2200 blend. Nonetheless, the ESA feature is unlikely to be directly contributed by the population occupied at defect states. Generally, the defect states can be populated by trapping the carriers created by initial photon absorption. In such case, the population dynamics of those defect states should exhibit a delayed rise following the optical pumping (e.g. Marciniak et al., Phys. Rev. B 85, 214204 (2012)). To relieve this concern, in the revised SI, we show the dynamics probed at 1.12 eV with a higher temporal resolution (Supplementary Figure 5). The ESA signal is built up simultaneously with the optical pumping, suggesting that it is related to an excited state populated by the initial optical absorption.

Reviewer’s Technical Comment #6:

Another point that the authors do not address is that from the very first TA map they report of the blend (400 fs), a bleach of the donor is present. As the authors showed

that there is no direct excitation of the donor with their 1.75 eV pump, this implies that substantial charge separation has already occurred by 400 fs!

Response:

Figure R6. (a) TA data from a neat J51 film recorded with pump at 1.75 eV. No signal is measured except the signal near zero time delay which is caused by coherent Stark effect in the polymer. (b) The dynamics probed at 2.05 eV in the J51/N2200 blend film recorded with pump at 1.75 eV. The rising curve can be fitted by an exponential growth function with a lifetime parameter of 3 ps (red curve). Coherent artifact is observed near zero time delay which is plausibly related to the Stark effect in the polymer.

We thank the reviewer for pointing out this issue. In the original manuscript, the temporal resolution of TA spectroscopy probed in the near infrared range is ~ 250 fs which cannot well resolve the dynamics of hole transfer at the early stage. To resolve this issue, we have performed new TA spectroscopy measurements with a higher temporal resolution of < 20 fs in the visible range and < 30 fs in the near infrared range using a non-collinear OPA as the pump source. As shown in Figure R6, the bleaching of the donor recorded at 2.05 eV has an ultrafast appearance within 30 fs which is entangled with the coherent stark effect (Becker et al., Phys. Rev. Lett. 60, 2462(1988)). This is probably related to the coherent charge transfer process as discussed in literature (Falke et al., Science 344, 1001 (2014)). Nonetheless, the major part of the signal is generated in the hole transfer process with a characterized lifetime of ~ 3 ps.

Figure R6 has been included as a new Supplementary Figure 2 in the revised SI. More discussions related to this issue are included on Page 7-8 in the revised manuscript and in Supplementary Note 1.

Reviewer's Technical Comment #7:

Further, the data from Figure 4 seems indeed strange. Are the authors sure the signal

they measure in the blend is not arising from a part of the film where mostly the acceptor is present? My feeling is that the data from Figure 4 might be representing inhomogeneity of the samples more than a fundamental process occurring dominantly. The hole transfer takes place on a timescale below the temporal resolution, so the authors only see signal of isolated acceptor domains that emit like the pristine. What determines the ratio of excitons and iPPs? My understanding is that iPPs do not absorb light directly, so if they are formed over time, why are there any excitons left behind to emit? Could the authors perform electroluminescence measurements or PL quantum yield to address this?

Response: We agree with the reviewer that the sample inhomogeneity should be carefully considered in analyzing the PL data (Figure 4). Following the reviewer's suggestion, we quantify the PL quantum yields in these samples which represent the global properties of the photo-induced excitons. Under 405 nm excitation, the quantum yields are $\sim 1\%$ and $\sim 0.35\%$ in the neat N2200 film and the blend J51/N2200 film, respectively. Considering only one-third of excitation light at 405 nm is absorbed by polymer acceptor N2200 in the blend (Figure R7(a)), the experimental results well support our assessment that the PL dynamics in N2200 are similar in the two samples as revealed by the TRFL spectra (Figure R7(b)). In addition, the morphology characterized by PiAFM shows that the donor and acceptor are well separated with small domain size of ~ 10 nm in the blend film optimized for device. The isolated domains of either donor or acceptor with large sizes that emit like the pristine should be rare.

Notably, although the temporal resolution may be insufficient to measure the extract rate of charge transfer, the quench of exciton fluorescence cause by charge transfer can be captured by TRFL traces. As an evidence FL dynamics of the donor emission show dramatic differences in the neat donor and blend films due to the exciton-mediated electron transfer from donor to acceptor (see Figure R7(c)). In another word, if the excitons in acceptor are largely involved in the hole transfer process, the difference in dynamical behaviors should manifest itself in TRFL spectra of the neat acceptor and blend films.

Figure R7. (a) Absorption spectrum of the blend film used in quantifying the PL quantum yields. The dashed and dot lines represent the absorptions contributed by the J71 and N2200 samples, respectively. (b) TRFL traces recorded at the emission peak of N2200 in the neat N2220 and the J51/N2200 blend films, respectively. (c) TRFL traces recorded at the emission peak of J51 in the neat J51 and J51/N2200 blend films, respectively.

Figure R8. The early-stage dynamics probed at 1.75 eV and 2.52 eV recorded with a higher temporal resolution. The oscillations imply that coherent vibronic coupling is possibly involved in the generations of iPPs and excitons.

Different scenarios have been proposed to explain the formation mechanism of iPPs in polymers. As noted by the reviewer, iPPs may be formed over time from excitons (e.g., Pensack & Asbury, *J. Phys. Chem. Lett.* 1, 2225 (2010); Rolczynski et al., *JACS* 134, 4142(2012)) or together with excitons simultaneously within the pulse duration of incident pulses (e.g., Guo et al., *J. Am. Chem. Soc.* 131, 16869 (2009); Tautz et al., *Nature Commun.* 3, 970 (2012) & etc.). Our experimental data suggest that the latter scenario is responsible for the formation of iPPs in polymer N2200. The

ESA signals for both iPPs and excitons are built up in an ultrafast temporal scale of less than 30 fs (see Figure R5 above). Such simultaneous formations of iPPs and excitons have been recently explained as a result of coherent vibronic coupling (De Sio et al., Nature Commun. 7, 13742 (2016)). The initial optical absorption creates a coherent state composed of vibronically-coupled iPPs and excitons which is converted to iPPs and excitons after the loss of coherence. Afterwards, the iPPs and excitons have different and uncorrelated dynamics like the results reported in literature (Tautz et al., Nature Commun. 3, 970 (2012)). The generation dynamics of iPPs is consistent with such initial coherent-state scenario. Primary data from broadband TA measurements (Figure R8) show certain oscillatory behaviors in the ESA signals of iPPs and GSB signals at the frequency of $\sim 1450 \text{ cm}^{-1}$, which is consistent with the coherent scenario for the iPP formation.

In the revised manuscript, we have included the above data in Supplementary Figure 7b and Supplementary Figure 5. The revisions related to the fluorescence measurements are made on the 1st Paragraph, Page 11 and Supplementary Note 2 in the revised SI. The revisions related to the generation dynamics are made on the 1st paragraph Page 17.

Reviewer's Technical Comment #8:

Finally, the authors do not perform similar experiments pumping the donor to characterize the electron transfer dynamics. From their figure 1c it seems like the most important contribution to device performance comes from electron transfer, and this data seems like a natural complement.

Response: We thank the reviewer for this valuable suggestion. We have now included the results of TA experiments on the electron transfer by pumping the donor as shown in Figure R9. Indeed, the electron transfer contributes significantly to the device performance, which can be observed with the increased decay rates of GSB and ESA features of the donor (Figure R9(b-d)) in the blend films. Due to the absorption overlap of the donor and acceptor in the spectral range of the pump pulse, it is challenging to clearly assign a spectral feature to the electron transfer. In spite of such difficulty, we can observe a significantly shortened PL lifetime of donor emission in TRFL traces (Figure R7c) in the blend in comparison with the neat donor as discussed above. The results clearly indicate that the photo-excited excitons are largely involved in the electron transfer process which is quite different from that in the case of hole transfer. Such a difference can explain the achieved higher external quantum efficiency in the spectral range of donor absorption in comparison with that of the acceptor absorption. We have included Figure R9 and Supplementary Figure 8 in the revised SI. The related discussion are made in Supplementary Note 2 and 3.

Figure R9. Electron transfer dynamics in the J51/N2200 blend. (a) TA spectra recorded from the blend film with pump at 2.04 eV. (b-d) TA traces probed at the GSB features at 2.04 eV and two ESA features at 1.12 eV and 1.31 eV, respectively.

Reviewer's Summary Comments:

In summary, I have great suspicion regarding the assignment of the iPP and exciton spectral features, which is the foundation for the discussion that follows. I also think that the paper is not very well written, lacking objectiveness and conciseness. The authors are repetitive a few times, wasting space that could be devoted to clearer explanations on more critical points. For instance, if the authors want to publish in a broad readership journal, at least a basic discussion to help readers from nearby fields to orient themselves on the iPP concept along with literature citations to orient readers. Perhaps this is already provided, but certainly not in a well organized and fluid way.

I think the paper cannot be published anywhere as it is, and could be reconsidered at Nature Communications following major revisions which address the points above.

Response: We greatly appreciate the reviewer's thoughtful review of the manuscript and the constructive suggestions. The reviewer's main concern on the assignments of the iPP and exciton spectral features might be caused by the missing of some key experimental data in the original manuscript, which we have now provided in responding to answer reviewer's comments. Reviewer's critical comments have

pushed us to carry out several additional experimental measurements, which provide more solid experimental evidences for our claims. We have addressed all the comments/concerns raised by the reviewer. Following the reviewer's suggestion, we have substantially modified the manuscript for a better introduction of the research background and provided better orientation for the broad readership. The changes are made in the introduction section on Page 3 and in the discussion section on Page 16-17 of the revised manuscript. In addition, we have consulted commercial company for proofreading to polished the language in the manuscript. With the substantial new experimental data and additional discussions in addressing the reviewer's comments, as well as the much modified presentation, the revised manuscript is significantly improved. Hopefully, the reviewer will now find the revised manuscript suitable for publication in Nature Communications.

Response to the Comments of Reviewer #2:

Reviewer's Comments:

The manuscript intended to suggest a new mechanism in all polymer solar cell which is distinctively different than polymer/PCBM solar cell. They investigated role of so called intra-moiety polaron pairs (iPPs) and concluded that these iPPs play an important role that was not present in polymer/fullerene solar cells. I am afraid that this conclusion is totally wrong and the authors lack the knowledge in fundamental photophysics of conjugated polymers. In particular, they have not fully understood the mechanisms revealed so far on so called charge transfer polymers or donor-acceptor polymers or low band gap polymers. There have been numerous literature since ~2005 (2005 PRL Friend and coworkers, 2008 JACS Durrant and coworkers, 2012 JACS Rolczynski et al., and many others) to show that the built-in charge transfer nature within a single polymer as well as their likely self-folding conformation promote the generation of charge transfer (CT) states after exciton splitting where the hole and electron are separated locally which is equivalent as the authors' iPPs. It has been suggested in the literature that the nature of such CT or iPP states in these polymers defray the exciton binding energy due to the hole-electron displacement along the polymer chain. Therefore, their work just proved such mechanism because they also used charge transfer polymer as the p-typed material. There is no fundamental difference between the all-polymer solar cells and polymer/fullerene solar cells. Their perceived "new" mechanism is a part of fundamental photophysics in these organic materials. It is my personal view that the main difference between the all polymer solar cell vs. polymer/fullerence solar cell will be their donor-acceptor morphology which have not been fully explored. Based on the above assessment, I disagree with authors view and hence will not recommend its publication in Nature

Communication due to the lack of novelty.

Response: The reviewer cites some papers in literature studying the photo-physics of iPP-like states in polymers to criticize the novelty of our work. It seems that the reviewer has misunderstood the key point in our manuscript. The main finding in our manuscript is not on the observation of iPPs in polymers. In this work, we focus on uncovering the roles of iPPs and excitons played during hole transfer in a high performing all-polymer J51/N2200 blend. We report compelling evidences that photo-induced iPPs, rather than photo-induced excitons, played a dominated role in triggering hole transfer process in this non-fullerene OPV blends. The iPPs are generated simultaneously together with the excitons in this prototypical polymer acceptor N2200 which mediates the hole transfer independently in J51/N2200 blends. The iPP-initiated process is markedly different from the scenario of excitons (EX) → interfacial charge-transfer states (xCT) → CS as intensively studied in polymer/fullerene systems.

The three papers mentioned by the reviewer either focus on different stages of charge generation process or simply cannot be found.

The first paper (“2005 PRL Friend and coworkers”) cannot be found from the database. There was no PRL paper from Friend’s group in 2005. Is it possible to be the PRB paper from the group (Phys. Rev. B 71, 24302 (2005)) where the iPP-liked state is mentioned? However, this paper has not reported any data about the role of iPPs during charge separation in OPV blends. We wish the exact citation was given by the reviewer.

For the second mentioned paper (“2008 JACS Durrant and coworkers”), there were four JACS papers from Durrant’s group in 2008. Only the JACS paper (titled “Charge carrier formation in polythiophene/fullerene blend films studied by transient absorption spectroscopy (Hideo et al., JACS 130, 3030 (2008))”) is related to the OPV mechanism. This paper reports the data supporting that free charges are generated from photo-induced excitons through an intermediate state of a Coulombically bound radical pair (BRP) (Schemes 1 & 2 in the paper). Such a BRP state is formed with electron and hole polarons at the acceptor and electron sites respectively after the initial stage of interfacial charge separation. The BRP state is actually similar to the xCT state in the conventional picture of exciton-initiated charge separation process in polymer/fullerene blends.

The third mentioned paper is titled “Ultrafast intramolecular exciton splitting dynamics in isolated low-band-gap polymers and their implications in photovoltaic materials design” from Chen’s group (Rolczynski et al., JACS 134, 4142 (2012)). This paper has studied the generation of different states with intramolecular CT nature from the photo-excited excitons, including valuable knowledge about the generation

mechanism of iPP-like states in polymers. We appreciate the reviewer for reminding us this paper and have cited it in the revised manuscript. Nonetheless, this paper doesn't weaken the novelty of our work. As noted in the first paragraph of our original manuscript, we have already introduced some previous studies on the generation of iPP-like states (Guo et al., JACS 131, 16869 (2009); Sheng et al., PRB 75, 085206 (2007); Tautz et al., Nature Commun. 3, 970(2012); JACS 135, 4282 (2013); & etc). The main finding in our manuscript is not on the observation of iPPs in polymers, but on the identification of the channel of iPPs instead of excitons as the major pathway for hole transfer in the J51/N2200 blend. The paper (Rolczynski et al., JACS 134, 4142 (2012)) focuses on neat polymers where two different types of iPP-liked states, namely "intramolecular pseudo-charge-transfer (PCT) states" and "intramolecular charge-separated states", have been identified. No experimental data are shown on how these states are involved in charge separation in OPV blends. We'd like to note that the iPP states observed in our work show significant different features from the iPP-liked "intramolecular PCT" and "intramolecular charge-separated" states (Rolczynski et al., JACS 134, 4142(2012)). The iPP-like states in that paper (Rolczynski et al., JACS 134, 4142(2012)) are generated from photo-induced excitons with delayed rises in the kinetic curves. In our work, the iPPs in the polymer acceptor N2200 are simultaneously generated together with excitons within the temporal resolution limit (see Figures R5 & R8 in the response to first reviewer), following the coherent scenario proposed recently (De Sio et al., Nature Commun. 7, 13742 (2016)).

In polymer/fullerene blends, charge generation process has been mainly explained with exciton-initiated process (i.e., exciton \rightarrow xCT \rightarrow CS) in the literature (Clarke & Durrant, Chem. Rev. 110, 6736 (2010)). It has been debated whether iPPs contribute substantially to or go against the charge separation in polymer/fullerene blends, although the generation of iPPs in some polymers has been observed for over a decade. In some polymers, iPP-liked states are generated over time from dissociation of EXs which may act as intermediate states for EX-initiated charge separation processes. It has been argued that "photogenerated PPs recombined too quickly (< 0.5 ns) to be extractable from the electrodes" (Di Nuzzo et al., JPC Lett. 6, 1196 (2015)). Similar argument has also been made in the paper noted by the reviewer (Rolczynski et al., JACS 134, 4142 (2012)) where the iPP-liked "PCT" state is a loss channel due to geminate recombination. In the blend systems with donor-acceptor polymers and fullerenes, the exciton-dominated processes have also been regarded as the major channels for the charge separation process as reported in recent works including several papers from the groups noted by the reviewer, such as Friend (Science 335, 1340 (2012); Nature Commun. 8, 277(2018)), Durrant (Nature

Commun. 9, 2059 (2018)), Chen (JPC Lett.5, 1856 (2014)) and co-workers.

We have experimentally shown that hole transfer in J51/N2200 blend is mainly contributed by the photo-excited iPPs while the excitons act unexpectedly as a spectator. Our work represents the very first study on the role of iPPs played in charge generation in non-fullerene all-polymer OPV materials. The iPP-dominated hole transfer process in the J51/N2200 blend is substantially different from the exciton-dominated electron transfer process in polymer/fullerene blends. N2200 is a prototypical non-fullerene polymer acceptor widely used in all-polymer devices. Efficient light energy harvested from hole transfer is a major advantage of non-fullerene OPV materials over the conventional polymer/fullerene systems as shown in previous works from us and several other groups (Bin et al., Nature Commun. 7, 13651 (2016); Stoltzfus et al., Chem. Rev. 116, 12920 (2016); Zhang et al., Chem. Rev. 118, 3447 (2018); & etc.). The hole transfer from N2200 to polymer donor has marked differences from the electron transfer process from polymer to fullerene acceptor. The morphology noted by the reviewer is surely an important factor for hole transfer process. Of equal importance are the differences in the charge withdrawing ability of polymer and fullerene, the localization of the photoexcited states, the density of the available accepting states, and the phonon modes involved (Long and Prezhdo, Nano Lett. 14, 3335 (2014)). Our finding not only sets a solid example that iPPs can contribute substantially to charge generation in all-polymer OPV devices but also uncovers the loss mechanism in J51/N2200 blend due to the lack of exciton-mediated hole transfer in comparison to OPV blends with small molecule non-fullerene acceptors. We have studied other blend films with polymer J51 and three different naphthalene diimide-based copolymer acceptors (Supplementary Figure 17). Similar iPP-mediated hole transfer process has been identified in all these blends, suggesting that the findings are quite general in all-polymer OPV systems.

To relieve the reviewer's concern and clarify the potential ambiguity, we included some additional discussions and cited some of the above mentioned papers for a better justification of the novelty of our finding. The related changes are made in 1st paragraph on Page 3 as "The iPP-liked ... photocurrent generation" and a whole paragraph of discussion on the differences between all-polymer and polymer/fullerene systems on Page16-18.

Response to the Comments by Reviewer #3:

Reviewer's General Comment:

The research topic is very interesting. It disclosed the hole transfer from the acceptor polymer to the donor polymer in all-PSCs is mediated by photo-excited polaron pairs, which is different from the well-established polymer/fullerene system. This study provided a deep understanding of all-polymer system and is meaningful for the further development of all-polymer systems. Few questions need to be addressed before considering for publishing.

Response: We sincerely appreciate that the reviewer found our manuscript “very interesting” and gave an accurate summary for the significance of the current work. We are very glad to see that the reviewer recommended it for publication after some proper revisions. In the following, we answer reviewer’s specific comments in detail and indicate changes made in the revised manuscript.

Reviewer's technical comment #1.

It will be very interesting to make a direction comparison to the polymer/fullerene system, for instance using J51/PCBM[70].

Response:

Figure R10. TA dynamics in a J51/PCBM[70] blend film under optical pump at 400 nm. (a) TA spectra recorded at different time delays. (b) The dynamics probed at 2.05 eV in the neat J51 and J51/PCBM blend films.

Following the reviewer’s suggestion, we have studied the interfacial hole dynamics in a blend film of J51/PCBM[70]. Due to spectral overlap, we pump the blend film at 400 nm to reduce the direct excitation of J51 as shown in Figure R10. We observe a feature of delayed rise of bleach signal in the J51 sample (see Figure 10(b)) which is probably related to hole transfer. Unlike the polymer N2200, excitons are the primary formation of the excited states in small-molecule acceptor PCBM. In the blend of J51/PCBM, hole transfer is primarily triggered by excitons, which is

significantly different from the iPP-mediate hole transfer process in the J51/N2200 blend. In this all-polymer system, iPPs are responsible for hole transfer while excitons in the acceptor are largely uninvolved.

We have included Figure R10 in the revised SI as Supplementary Figure 16 and related discussion in Supplementary Note 6.

Reviewer's technical comment #2. *It is hard to draw a general conclusion on all-PSCs based on only one material system. It will make the study more interesting by extending to another polymer acceptor system.*

Response:

Figure R11. iPP-mediated hole transfer process in all-polymer blends with different polymer acceptors. (a) Molecule structures of donor-acceptor co-polymer acceptors. (b) Representative TA spectra from J51:P(IDT-NDI) blend films recorded at different time delays. (c-e) Polaron induced absorption change in different polymer acceptors. (f-h) Dynamics of iPPs in the neat acceptor and blend films.

This is a very good suggestion. To draw a general conclusion, we have studied

more blend systems using the same polymer donor J51 and three different polymer acceptors of donor-acceptor copolymers, i.e., P(IDT-NDI), J51/P(TVT-NDI) and J51/P(NDI2DT-T). In all three systems, the iPP-mediated hole transfer process has been observed as shown in Figure R11. Similar polaron pair-mediated hole transfer processes have been identified in all those three blends, indicating that our findings are quite general in all-polymer systems. These newly generated results establish wider implications of our findings in all-polymer OPV systems.

We added the Figure R11 to the revised SI as Supplementary Figure 17. The relevant discussions are included on Page 17-18 as “We have... all-polymer OPV systems” in the revised manuscript and Supplementary Note 6 in the revised SI. Supported by the experimental data, we have changed the words “an all-polymer photovoltaic blend” to “all-polymer photovoltaic blends”.

Reviewer’s technical comment #3.

When the morphology changes, how it will influence the quantum efficiency of hole transfer? The numbers need to be provided.

Response: In the ultrafast temporal scale (< 10 ps), the hole transfer process is mainly contributed by the photo-excited iPPs. The quantum efficiency of iPPs that undergo hole transfer is ~ 97 % in the samples annealed at temperatures below 110 °C, but drops to ~ 83 % in the sample annealed at 200 °C. The morphology effect is also manifested as different quantum efficiencies of iPP generation in samples annealed at different temperatures. Following the established approach in literature (e.g., Tautz et al., Nature Commun. 3, 970 (2012)), we quantify the efficiency of iPP generation. The iPP yields in the blend samples annealed at temperatures of 70 °C, 110 °C, 150 °C and 200 °C are estimated be 39%, 43%, 30% and 26%, respectively.

These numbers are reported in the revised Figure 5 in the main manuscript and more calculation details are added in Supplementary Note 4 in the revised SI. The related discussions are included as “The quantum efficiency... at 200 °C” and “We quantify the generation...all-polymer blends” on Page 13-14.

Reviewer’s technical Comment #4.

If the hole transfer is so efficient, how about the internal quantum efficiency? What is the main loss mechanism since the EQE is not so high?

Response: This is a good question. The efficiency of a OPV device is sensitive to many factors besides the process of ultrafast charge generation studied in this work. The EQE of the all-polymer device is not so high in particularly for the spectral range of acceptor absorption, which is possibly related to the inefficient hole transfer from excitons in the J51/N2200 blend in the view of charge generation dynamics. Optical

absorption creates both iPPs and excitons in N2200. As discussed in the manuscript, the iPPs in the acceptor can contribute to the hole transfer process at a markedly high efficiency, but the generated excitons are less involved in the ultrafast hole transfer process. Such a loss channel, together with the relatively weak absorption of N2200, limits the EQE in the spectral range of acceptor absorption. To avoid such loss, new strategies need to be developed to improve the hole transfer from photo-excited excitons.

We have included the related discussion “Notably, the spectator role...acceptor absorption” on Page 11 in the revised manuscript.

In summary, by providing additional experimental data and modifying the presentation in the revised manuscript and SI, we have addressed all the reviewers’ comments. With these revisions, the conclusions are strongly supported and the wider implications of the major finding are well justified. Thus, we strongly believe that the manuscript is suitable for publication in Nature Communications.

REVIEWERS' COMMENTS:

Reviewer #1 (Remarks to the Author):

The authors have deeply revised the manuscript by performing both new experiments (spectroelectrochemistry and high time resolution transient absorption, experiments on different donor-acceptor blends) as well as data analysis and have put their conclusions on much firmer ground.

I believe that my objections have been satisfactorily addressed and the paper is now ready for publication in Nature Communications.

Reviewer #3 (Remarks to the Author):

The authors have done a lot of extra experiments and all my comments were well addressed. I am satisfied with the current version.